# Safety of Tacrolimus Monotherapy within 12 Months after Liver Transplantation in the Era of Reduced Tacrolimus and Mycophenolate Mofetil: National Registry Study

**DOI:** 10.3390/jcm11102806

**Published:** 2022-05-17

**Authors:** Deok Gie Kim, Sung Hwa Kim, Shin Hwang, Suk Kyun Hong, Je Ho Ryu, Bong-Wan Kim, Young Kyoung You, Donglak Choi, Dong-Sik Kim, Yang Won Nah, Jai Young Cho, Tae-Seok Kim, Geun Hong, Dong Jin Joo, Myoung Soo Kim, Jong Man Kim, Jae Geun Lee

**Affiliations:** 1Department of Surgery, Yonsei University College of Medicine, Seoul 03722, Korea; mppl01@yuhs.ac (D.G.K.); djjoo@yuhs.ac (D.J.J.); ysms91@yuhs.ac (M.S.K.); 2Department of Biostatistics, Yonsei University Wonju College of Medicine, Wonju 26426, Korea; juniver1057@naver.com; 3Department of Surgery, Asan Medical Center, University of Ulsan College of Medicine, Seoul 05505, Korea; shwang@amc.seoul.kr; 4Department of Surgery, Seoul National University College of Medicine, Seoul 03080, Korea; nobel1210@naver.com; 5Department of Surgery, Pusan National University Yangsan Hospital, Pusan National University School of Medicine, Busan 49241, Korea; ryujhhim@hanmail.net; 6Department of Liver Transplantation and Hepatobiliary Surgery, Ajou University School of Medicine, Suwon 16499, Korea; drbwkim@ajou.ac.kr; 7Department of Surgery, College of Medicine, The Catholic University of Korea, Seoul 06591, Korea; yky602@catholic.ac.kr; 8Department of Surgery, Catholic University of Daegu, Daegu 42472, Korea; dnchoi@cu.ac.kr; 9Department of Surgery, Korea University College of Medicine, Seoul 02841, Korea; kimds1@korea.ac.kr; 10Department of Surgery, Ulsan University Hospital, University of Ulsan College of Medicine, Ulsan 44033, Korea; nahyw@uuh.ulsan.kr; 11Department of Surgery, Seoul National University College of Medicine, Bundang Hospital, Seongnam 13620, Korea; jychogs@gmail.com; 12Department of Surgery, Dongsan Medical Center, Keimyung University School of Medicine, Daegu 42601, Korea; gskim80094@naver.com; 13Department of Surgery, EWHA Womans University College of Medicine, Seoul 07804, Korea; ltdrhong@gmail.com; 14Department of Surgery, Samsung Medical Center, Sungkyunkwan University School of Medicine, Seoul 06531, Korea

**Keywords:** liver transplantation, tacrolimus, mycophenolate mofetil, renal dysfunction, time-conditional propensity score

## Abstract

Tacrolimus monotherapy is accepted as a feasible option during early post-liver transplantation as per current international consensus guidelines. However, its effects in the recent era of reduced tacrolimus (TAC) and mycophenolate mofetil (MMF) remain unclear. Liver recipients who either received TAC monotherapy from the treatment onset or switched from TAC/MMF to TAC-mono within 12 months (TAC-mono group; n = 991) were chronologically matched to patients who continued to receive TAC/MMF (TAC/MMF group; n = 991) at the corresponding time points on time-conditional propensity scores. Outcomes within 12 months after matched time points were compared. Biopsy-proven rejection (TAC/MMF: 3.5% vs. TAC-mono: 2.6%; *p* = 0.381) and graft failure (0.2% vs. 0.7%; *p* = 0.082) were similar in both groups. However, the decline in eGFR was 3.1 mL/min/1.73 m^2^ (95% CI: 0.8–5.3) greater at six months (*p* = 0.008) and 2.4 mL/min/1.73 m^2^ (95% CI: −0.05–4.9) greater at 12 months (*p* = 0.048) after the matched time points in TAC-mono group than that in TAC/MMF group. TAC trough levels were also higher in the TAC-mono group throughout the study period. TAC-mono within 12 months after liver transplantation is immunologically safe. However, it can increase the required TAC dose and the decline in renal function than that in TAC/MMF combination therapy.

## 1. Introduction

As survival after liver transplantation (LT) continues to improve [1], increasing emphasis is being placed on the long-term outcomes of transplant recipients. Optimization of immunosuppressive therapy is one of the most important factors for preventing long-term complications after LT [2]. Tacrolimus (TAC) is universally used in LT to prevent rejection, although its use is associated with long-term complications, including renal dysfunction [3] and increased risks of malignancy [4] and diabetes mellitus [5]. To reduce TAC exposure, mycophenolate mofetil (MMF), a T- and B-cell specific immunosuppressant [6], is frequently combined with TAC [1]. Compared with standard-dose TAC, the combination of MMF and reduced-dose TAC decreases renal dysfunction without increasing graft rejection [7]. However, MMF is associated with its own set of adverse events, such as gastrointestinal disturbance [8], leukopenia, thrombocytopenia [7], and teratogenicity [9], which can require avoidance or discontinuation of the drug. A study conducted in the United States reported that although 68% of patients were receiving TAC/MMF combination therapy at hospital discharge after LT, the proportions of patients receiving TAC monotherapy or TAC/MMF (with or without glucocorticoids) were both approximately 40% at one year after LT [10].

Recent International Liver Transplantation Society guidelines suggested that TAC monotherapy is feasible after three months post-LT in patients at low risk for rejection [11]. However, this recommendation was based on a study performed over a decade ago, which compared TAC monotherapy with TAC/glucocorticoid combination therapy after LT; MMF was not included in the study protocol [12]. Research comparing avoidance or discontinuation of MMF with TAC/MMF combination in LT is lacking. Using a national multicenter database, we conducted this study to compare the safety of TAC monotherapy versus TAC/MMF combination therapy in the first 12 months after LT.

## 2. Materials and Methods

### 2.1. Study Population

For this retrospective cohort study, we used data from the Korean Organ Transplantation Registry (KOTRY) of patients who underwent LT surgery between April 2014 and December 2019. The structure and methods of KOTRY have been previously described [13]. We identified 3434 patients who underwent LT during this time period. The exclusion criteria from baseline characteristics were age under 19 years (n = 115), retransplantation (n = 51), other malignancy than HCC detected on explant pathology (n = 97), and dual-graft living donor LT (n = 49). To assess the immunologic safety of TAC monotherapy, we also excluded 165 patients with immune-mediated liver diseases (e.g., autoimmune hepatitis, primary biliary cirrhosis, primary sclerosing cholangitis), as these disorders increase the risk of rejection [14,15]. Dual-graft LDLT (n = 49) and lack of data (n = 206) After applying these exclusion criteria, 3087 patients were eligible for the base cohort in this study.

### 2.2. Data Collection

KOTRY contains baseline characteristics of transplant recipients and donors, as well as laboratory values for aspartate aminotransferase (AST), alanine aminotransferase (ALT), total bilirubin, and serum creatinine obtained from at one, six, and 12 months after LT, and then annually. Data regarding the use of each immunosuppressant agent and TAC through serum levels were also collected at the same time. We calculated the estimated glomerular filtration rate (eGFR) from the serum creatinine using the Modification of Diet in Renal Disease equation [16]. The performance status of each patient after LT was categorized according to the Karnofsky performance status score as high (80–100%), intermediate (50–70%), or low (0–40%).

### 2.3. Study Design

To account for the limitations of a retrospective study, we performed matched analysis on the time-conditional propensity score (Figure 1), which is a well-described pharmacoepidemiologic design [17]. The TAC monotherapy group included patients treated with TAC monotherapy from one month after LT and those in whom the immunosuppressant regimen was switched from TAC/MMF to TAC monotherapy at six or 12 months after LT. The TAC/MMF comparator group included patients maintained on TAC/MMF at the same time points (one, six, or 12 months). Patients treated with a drug regimen other than TAC/MMF before the given time points were not included in the comparator group. Exclusion criteria were applied for time-varying characteristics in each time-based matching cohort, death or loss to follow up within three months after the matched time of exposure, prior biopsy-proven rejection (BPR), prior recurrence of HCC, more than five times the upper limit of normal in liver function tests (AST, ALT, and total bilirubin), and invalid time-varying variables.

We then matched the TAC/MMF group to the TAC monotherapy group in each time-based cohort (one, six, and 12 months) using time-conditional propensity scores generated via a conditional logistic regression, which included all baseline- and time-varying variables measured at the corresponding time points [18]. In addition to the Karnofsky performance status, glucocorticoid use, liver function results, and eGFR, infectious complications were also included in the calculation of the propensity score to ensure greater equity between the groups. Matching was performed in chronological order (i.e., one month, six months, 12 months). Patients in the TAC monotherapy group were matched to those in the TAC/MMF group at a ratio of 1:1. Patients in the TAC/MMF group who had been selected as matched comparators once were not considered as possible comparators in subsequent matching. If a patient in the matched TAC/MMF group switched to TAC monotherapy during follow-up, the data at the time of the switch were censored, and the case was included in the TAC monotherapy group in the next matching set. Furthermore, the matched cases from both groups were censored if the regimen changed during follow-up. The primary endpoint of this study was BPR, which was defined as any rejection reported from a liver biopsy. Follow up period was until death or 12 months after matched time points whichever came first owing to compare short-term immunological outcomes resulting from changes of immunosuppressants. The secondary endpoints were graft failure, which was defined as receiving retransplantation or patient death from graft dysfunction and change in eGFR within 12 months after the matched time point.

### 2.4. Statistical Analysis

Matching on time-conditional propensity scores was performed using the nearest neighbor method with a caliper of 0.2 standard deviations. The balance of covariates was confirmed using standardized mean differences, with values between −0.1 and 0.1 considered adequate matching [19]. During the matching process, patients outside of balance were discarded in both groups.

Categorical variables were presented as numbers (proportion) and compared between matched groups using the chi-square test. Continuous variables were presented as the mean ± standard deviation or median (interquartile range [IQR]) and compared with the Student *t*-test or Mann-Whitney U test, depending on the normality of the covariates. Survival outcomes were compared from matched time points using Kaplan-Meier analysis and the log-rank test. All analyses were performed using standard software (SPSS v25.0 [IBM, Armonk, NY, USA] and R freeware v3.6.3 [R Foundation for Statistical Computing, Vienna, Austria]). *p* < 0.05 was considered statistically significant.

## 3. Results

### 3.1. Baseline Characteristics

After chronological matching based on the time-conditional propensity score, the final matched set included 991 patients in both the TAC/MMF and TAC-mono groups (details in Appendix A). All matching variables were confirmed to be in an adequate balance, as shown in Appendix A.

Table 1 shows the baseline characteristics of the matched population. There were no significant differences in variables between the two matched groups. The mean age was approximately 54 years, and there was a male predominance in both groups. Living related donors accounted for 68.1% of transplants in the TAC/MMF group and 70.3% of those in the TAC-mono group. The mean donor age was approximately 35 years, and there was a male donor predominance in both groups. ABO incompatibility was 13.7% in the TAC/MMF group and 13.8% in the TAC-mono group. The proportions of patients with pre-transplantation hypertension (17.7% vs. 17.4% in the TAC/MMF and TAC-mono group, respectively) and diabetes mellitus (26.3% vs. 26.0%) were also similar between groups. The most frequent underlying liver disease was hepatitis B (54.7% vs. 56.9%), followed by alcoholic liver disease (31.0% vs. 25.4%) in both groups. The pre-transplantation model for end-stage liver disease score was 17.6 ± 10.4 in the TAC/MMF group and 17.6 ± 10.0 in the TAC-mono group. The proportions of patients who had HCC within and above Milan criteria were 36.7% and 9.3% in the TAC/MMF group, whereas they were 36.8% and 8.1% in the TAC-mono group, respectively. The Karnofsky performance status, proportion of steroid use, liver function test results (AST, ALT, and total bilirubin), and eGFR at matched time points were similar between the groups. Rates of infectious complications before the matched time points were also similar between the TAC/MMF and TAC-mono groups (22.9% vs. 23.1%).

### 3.2. BPR and Other Survival Outcomes

As shown in Figure 2, within 12 months after the matched time points, the cumulative incidence of BPR was 3.5% in the TAC/MMF group and 2.6% in the TAC-mono group, with no significant difference between the groups (*p* = 0.381). The cumulative incidence of graft failure was also not significantly different between the groups (0.2% in the TAC/MMF group and 0.7% in the TAC-mono group; *p* = 0.082). The two groups also exhibited no significant differences in rates of patient death (0.7 vs. 1.3%; *p* = 0.075) or infection (11.4% vs. 14.4%; *p* = 0.079).

### 3.3. Renal Function

Figure 3 shows a comparison of the mean eGFR decline six and 12 months after the matched time points. After six months, the mean eGFR decline was −6.9 mL/min/1.73 m^2^ (IQR: −8.5 to −5.4) in the TAC/MMF group and −10.0 mL/min/1.73 m^2^ (IQR: −11.6 to −8.4) in the TAC-mono group. The inter-group difference in eGFR decline was 3.1 mL/min/1.73 m^2^ (95% confidence interval [CI]: 0.8 to 5.3), favoring the TAC/MMF group (*p* = 0.008). After 12 months from the matched time points, the median eGFR decline was −7.7 mL/min/1.73 m^2^ (IQR −9.4 to −5.9) in the TAC/MMF group and −10.1 mL/min/1.73 m^2^ (IQR: −11.9 to −8.3) in the TAC-mono group. The inter-group difference in eGFR decline was 2.4 mL/min/1.73 m^2^ (95% CI: −0.05 to 4.9), favoring the TAC/MMF group (*p* = 0.048) at this point as well.

### 3.4. Graft Function and Tacrolimus Trough Levels

Based on the results of AST, ALT, and total bilirubin tests, liver graft function was similar between the groups throughout the study period (Figure 4a–c). However, median trough levels of TAC were lower in the TAC/MMF group than in the TAC-mono group at matched time points (6.7 ng/dL vs. 7.4 ng/dL; *p* < 0.001), at six month (5.7 ng/dL vs. 6.5 ng/dL; *p* < 0.001), and at 12 months later (5.4 ng/dL vs. 5.7 ng/dL; *p* = 0.050, Figure 4d), all of which were within reduced-dose range in both groups.

## 4. Discussion

This study, using national registry data, compared the safety of TAC monotherapy with TAC/MMF combination therapy within the first 12 months after LT in patients with a low risk of rejection. In the matched analysis on time-conditional propensity scores, patients who received TAC monotherapy had similar BPR and graft failure compared with those who continued TAC/MMF. However, the TAC-mono group exhibited more decline in eGFR over the first 12 months after initiation of TAC monotherapy compared with that of the TAC/MMF group. We also found that TAC trough levels were significantly higher after discontinuing MMF in the TAC-mono group than in the TAC/MMF group. Our results, therefore, indicate that TAC monotherapy in the first 12 months after LT is immunologically safe, but it is associated with greater eGFR decline than TAC/MMF combination therapy, which may be due to higher trough levels of TAC.

In addition to studies published before 2000 [20,21,22], a large study performed in the modern era with improved immunosuppressive regimens also showed that rejection significantly increased graft failure or death in LT recipients [14]. Although there has been considerable interest in minimizing or even completely withdrawing immunosuppressants [23,24], the 10–30% incidence of rejection limits the application of these strategies in clinical settings [25,26]. A recent randomized trial reported that immunosuppressant minimization was achievable only in selected patients and did not result in reductions in infection, malignancy, or renal impairment [27]. In this context, our results provide useful evidence to help direct the tailored use of immunosuppressants.

TAC/MMF combination therapy, with or without glucocorticoids, remains the most popular immunosuppressant regimen for LT. A randomized trial showed that early conversion from TAC to everolimus (EVR) protected renal function [28,29]. However, the researchers also reported more BPRs and serious adverse events in the EVR conversion group, which could limit the utility of this TAC-withdrawal regimen, especially in the initial post-LT period. Recent studies have reported superior renal function and hepatocellular carcinoma outcomes when combining reduced doses of TAC with EVR or sirolimus [30,31,32,33], but the control groups in these studies received conventional TAC regimens, not reduced doses of TAC with MMF. Further research is required to determine whether EVR is preferable to MMF.

The use of TAC monotherapy due to adverse effects of MMF, such as bone marrow suppression and gastrointestinal problems [34], remains common in the clinical management of LT. However, the safety of TAC monotherapy vs. combination therapy with TAC/MMF and their possible effects on LT outcomes have not been established in this context. This subject is not eligible for investigation in a randomized controlled study, highlighting the need for well-designed retrospective studies. Nonetheless, retrospective medication studies may be limited by various time-related biases [35] and medication changes during follow-up. In this study, we reduced such biases by a systematic matching process using time-conditional propensity scores, allowing us to reasonably compare outcomes in TAC-mono and TAC/MMF groups that included patients with similar underlying conditions and time-varying factors, such as liver function, renal function, and prior infection at corresponding time points for use of each regimen.

In this study, we regarded glucocorticoid use as one of the matching variables, rather than part of the immunosuppressive regimen. Lerut et al. previously reported that glucocorticoid discontinuation three months after LT was feasible, even before the era of MMF [12]. The feasibility of steroid-sparing regimens has been more assured since the introduction of MMF in the field of LT [36]. Although one systematic review indicated the necessity for further evidence [37], current trends in immunosuppressive regimens after LT highlight how many clinicians accept the feasibility of glucocorticoid withdrawal: approximately 20% of LT patients in the United States received TAC/MMF without glucocorticoids even at the time of LT [1], and only 30% of LT patients in the KOTRY database were taking glucocorticoids by six months after LT (Appendix A). As our study population had a low risk of rejection, we assumed that maintenance glucocorticoids had minimal effect on immunologic outcomes. Therefore, we chose to control for their use during the matching process rather than include it in the immunosuppressive regimen.

The 2018 International Liver Transplantation Society guidelines suggest that TAC monotherapy is safe after three months post-LT based on a study published in 2008, which evaluated only glucocorticoid use, not MMF use [11]. The feasibility of TAC monotherapy requires re-evaluation in the current LT era, in which TAC/MMF is the mainstay of immunosuppressant therapy. In the current study, TAC trough levels tended to be higher after initiation of TAC monotherapy than when maintaining TAC/MMF combination therapy. Accordingly, the eGFR at six and 12 months after initiating TAC monotherapy was significantly lower than in the TAC/MMF group. This suggests that increasing the TAC trough level to prevent BPR with TAC monotherapy led to more decline in renal function although both groups showed a reduced-dose range of TAC throughout the study period. Thus, we recommend that guidelines for TAC monotherapy after LT consider individual risks for renal impairment, as well as BPR. Further study is required to determine the eligible standard and optimal TAC trough level for TAC monotherapy or the necessity for other additional immunosuppressants for patients in whom MMF discontinuation is necessary because of the development of or susceptibility to MMF adverse events.

This registry study has some limitations. For example, data collected regarding the times of immunosuppressive agent changes were based on the date of data collection rather than the actual time of the change. Moreover, policies for immunosuppressant usage varied among institutions contributing to KOTRY; therefore, the exact cause of MMF discontinuation could not be determined. Lastly, there may have been unmeasured confounding factors that differed between TAC-mono and TAC/MMF groups despite the successful matching.

## 5. Conclusions

In a matched analysis based on time-conditional propensity scores, this national cohort study showed that TAC monotherapy within the first 12 months after LT is immunologically safe. However, the higher dosage of TAC (although within a reduced-dose range) required to achieve this can lead to more decline in renal function than continued use of TAC/MMF combination therapy. These results highlight the need for more refined and personalized use of immunosuppressive agents in terms of not an only immunologic risk but also renal impairment when LT patients require avoidance or discontinuation of MMF under the priority of TAC/MMF combination therapy.

## Figures and Tables

**Figure 1 jcm-11-02806-f001:**
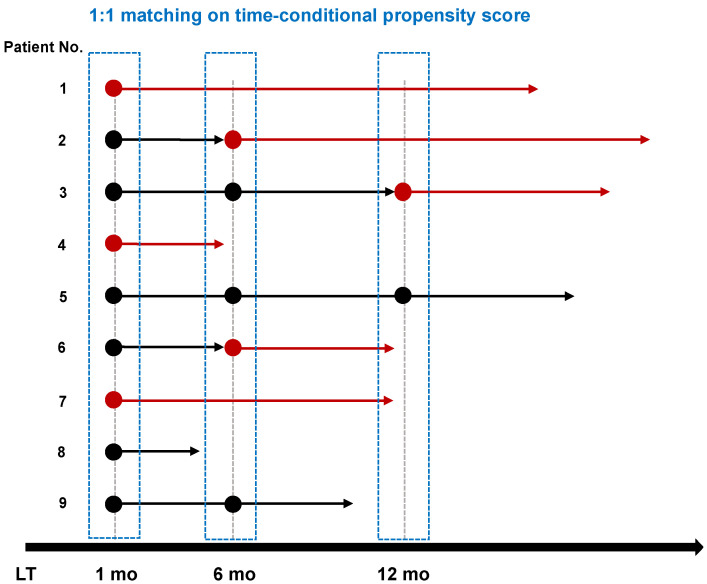
Graphical description of the time-based cohort for matching on the time-conditional propensity score. Round marks mean potential subjects for matching at each time point and arrows mean their follow-up. Black is for TAC/MMF combination therapy and red is for TAC monotherapy. TAC monotherapy users were matched to TAC/MMF users at a 1:1 ratio with time-conditional propensity scores which were generated with baseline- and time-varying variables measured at each time point, including Karnofsky performance status score, glucocorticoid use, liver function tests, eGFR, and infection before matched time points. TAC/MMF users who were selected as matched comparators for given TAC monotherapy users once were not considered as possible comparators in the subsequent matching process. Matched TAC/MMF users who changed to TAC monotherapy during follow-up were censored at the time of switch and included as a TAC monotherapy group in the next matching set. Both matched groups were censored if the regimen changed to another one during follow-up.

**Figure 2 jcm-11-02806-f002:**
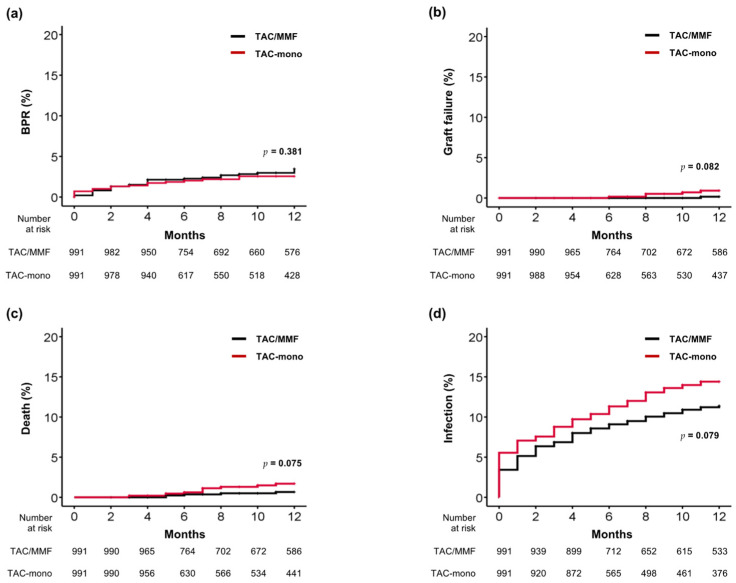
Comparison of outcomes between matched TAC/MMF and TAC-mono groups. (**a**) Biopsy-proven rejection, (**b**) graft failure, (**c**) death, and (**d**) infection. BPR, biopsy-proven rejection; MMF, mycophenolate mofetil; TAC, tacrolimus.

**Figure 3 jcm-11-02806-f003:**
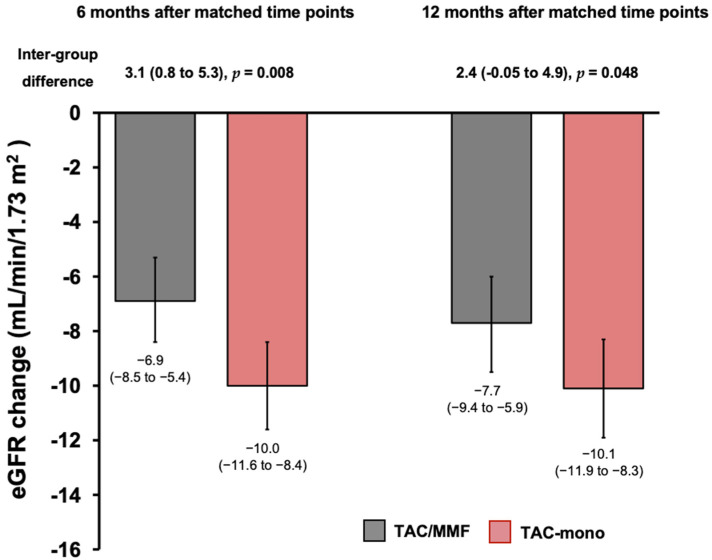
Comparisons between TAC/MMF and TAC-mono groups of the mean estimated glomerular filtration rate (eGFR) declines over six and 12 months after matched time points. The end of the box means mean eGFR decline and the error bar means a 95% confidence interval. LT, liver transplantation; MMF, mycophenolate mofetil; TAC, tacrolimus.

**Figure 4 jcm-11-02806-f004:**
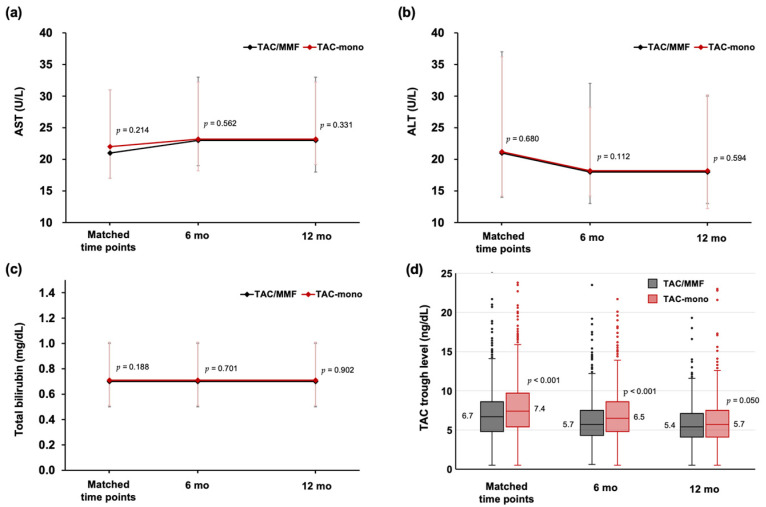
Comparison of liver function tests and tacrolimus trough levels between TAC/MMF and TAC-mono groups. Values of (**a**) AST, (**b**) ALT, (**c**) total bilirubin, (**d**) TAC trough level were compared at matched time points, six months and 12 months later. In the box-and-whisker plot (Figure 4d), horizontal lines in the boxes represent the median, and the box limits represent the interquartile range. The ends of the whiskers mean maximum and minimum values and dots mean outliers. MMF, mycophenolate mofetil; TAC, tacrolimus.

**Table 1 jcm-11-02806-t001:** Baseline characteristics in matched population.

Variables	TAC/MPA (n = 991)	TAC-Mono (n = 991)	*p*
Age, year	54.3 ± 8.7	54.4 ± 8.9	0.881
Sex, female	310 (31.3)	304 (30.9)	0.772
Body mass index, kg/m^2^	24.0 ± 3.6	23.9 ± 3.7	0.325
Year of LT			0.997
2014–2016	465 (46.9)	465 (46.9)	
2017–2018	526 (53.1)	526 (53.1)	
Donor type			0.406
Living related	675 (68.1)	697 (70.3)	
Living unrelated	112 (11.3)	95 (9.6)	
Deceased	204 (20.6)	199 (20.1)	
Donor age, year	35.6 ± 14.1	35.3 ± 14.1	0.707
Donor sex, female	359 (36.2)	372 (37.5)	0.550
ABO incompatibility	136 (13.7)	137 (13.8)	0.952
Hypertension	175 (17.7)	172 (17.4)	0.861
Pre-transplant DM	261 (26.3)	258 (26.0)	0.884
Underlying liver disease			0.068
Alcoholic	307 (31.0)	252 (25.4)	
HBV	542 (54.7)	564 (56.9)	
HCV	57 (5.8)	65 (6.6)	
Cryptogenic	56 (5.7)	69 (7.0)	
Drug	11 (1.1)	11 (1.1)	
Others	18 (1.8)	30 (3.0)	
Pre-transplant MELD	17.6 ± 10.4	17.6 ± 10.0	0.942
Pre-transplant HCC			0.623
No HCC	535 (54.0)	546 (55.1)	
Within Milan HCC	364 (36.7)	365 (36.8)	
Above Milan HCC	92 (9.3)	80 (8.1)	
Time-varying variables at matched time points			
Karnofsky performance status score			0.694
High (80–100%)	405 (40.9)	422 (42.6)	
Intermediate (50–70%)	501 (50.6)	482 (48.6)	
Low (0–40%)	85 (8.6)	87 (8.8)	
Steroid use	664 (67.0)	653 (65.9)	0.601
Liver function test			
AST, U/L	21 (17–31)	22 (17–31)	0.245
ALT, U/L	21 (14–37)	21 (14–36)	0.687
Total bilirubin, mg/dL	0.7 (0.5–1.0)	0.7 (0.5–1.0)	0.187
eGFR, mL/min/1.73 m^2^	78.8 ± 30.4	78.7 ± 32.9	0.920
Previous Infections	227 (22.9)	229 (23.1)	0.920

Data are presented by numbers (percentage) or mean ± SD. ALT, alanine aminotransferase; AST, aspartate aminotransferase; DM, diabetes mellitus; eGFR, estimated glomerular filtration rate; HBV, hepatitis B virus; HCC, hepatocellular carcinoma; HCV, hepatitis C virus; LT, liver transplantation; MELD, model for end-stage liver disease; MPA, mycophenoleic acid; TAC, tacrolimus.

## Data Availability

The data that support the findings of this study are available from Korean Organ Transplantation Registry (KOTRY), but restrictions apply to the availability of these data, which were used under license for the current study, and so are not publicly available. Data are however available from the authors upon reasonable request and with permission of KOTRY.

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
