# Peer review of "Safety of Tacrolimus Monotherapy within 12 Months after Liver Transplantation in the Era of Reduced Tacrolimus and Mycophenolate Mofetil: National Registry Study"

_jcm, 2022, doi:10.3390/jcm11102806_

Round 1

Reviewer 1 Report

I read with interest the paper evaluating the role of tacrolimus versus TAC/MMF.

Some points:

  • Why did you use 12 months as a follow-up, considering that most of the complications are by definition long term?
  • Did you consider any other possible causes of the decline of renal function after the LT?

Minor point:

  • Why did you choose 19 as limit age
  • The color in figure 3 is not corresponding with the legend

Author Response

Response to reviewer #1

Why did you use 12 months as a follow-up, considering that most of the complications are by definition long term?

=> We chose short-term follow up as 12 months after matched time points because the primary end point of this was biopsy proven rejection which should be compared within short-term after alteration of immunosuppressants. Another reason was that considerable number of patients had changed their immunosuppressants during follow up so long term follow up would result in heterogeneity of drug usage after matched time points. We added short explanation about this in the method section.

<Page 4>

Follow up period was until death or 12 months after matched time points which ever came first owing to compare short-term immunological outcomes resulted from changes of immunosuppressants.

Did you consider any other possible causes of the decline of renal function after the LT?

=> Yes, we matched all baseline and time varying variables at matched time points so TAC/MMF and TAC-mono group could have similar underlying conditions related with subsequent renal dysfunction. All about those method was described in our article.

Why did you choose 19 as limit age

=> We intended to include adults only as prior studies were generally performed

The color in figure 3 is not corresponding with the legend

=> Sorry, but we could not find any errors of colors in figure 3. Otherwise, please specify which part is incorrect.

Reviewer 2 Report

The authors assess the safety of tacrolimus therapy (withoit mycophenolate mofetil) in a large registry-based cohort of liver transplant recipients in South Korea.
The idea is excellent, the statistical method used is solid, the work is well written and I think the message would be interesting for the readers.

However, there are several points that I think the authors should clarify:

-there is a contradiction between the abstract and the results section: the values between graft failure and BPR rates are inverted
-it is unclear why patients with autoimmune conditions were excluded from the study
-a significant amount of patients transplanted for HCC were outside Milan criteria. Please state the criteria used in South Korea for HCC listing.
-sarcopenia is not used as a criteria used for matching, nevertheless it influences all the outcomes studied, especially renal function evolution
-in the methods section there are two definitions lacking: biopsy proven rejection (any severity? T-cell and antibody-mediated?) and graft failure (needing retransplantation? bili above a certain threshold? recurrent ascites?)
-the fact that the reason for MMF non-utlilization or discontinuation is missing, as stated in the discussion. This is one of the most notable weaknesses of the study, since it can lead to confusion bias? For example, if diarrhea was a frequent reason, then some patients in the TAC alone group could have suffered of dehydration caused by the diarrhea, thus impacting the renal function.
-there are typos in the text: page 2 line 80, all the citations in the Discussion section
-another limitation is the generalizability of the study: the cohort is pretty unique and difficult to compare to US or european LT cohorts since LDLT and HBV are overrepresented. This militation should be stated in the discussion section.

Author Response

Response to reviewer #2

-there is a contradiction between the abstract and the results section: the values between graft failure and BPR rates are inverted

=> Thanks for your kind review. We corrected the mistakes in the abstract as below.

< In the Abstract >

 Outcomes within 12 months after matched time points were compared. Biopsy-proven rejection (TAC/MMF: 3.5% vs. TAC-mono: 2.6%; P=0.381) and graft failure (0.2% vs. 0.7%; P=0.082) were similar in both groups.

-a significant amount of patients transplanted for HCC were outside Milan criteria. Please state the criteria used in South Korea for HCC listing.

=> Thanks for your opinion. In Korea, HCC patients within Milan criteria get exceptional MELD points according to their MELD at the listing (+4 for MELD 0~13, +5 for MELD 14~20, being at 25 points for MELD 21~25). There is no absolute prohibition of listing for above Milan HCC patients. However, as seen in our data, majority of study populations were LDLT patients because of Korea have been suffering from severe donor shortage. Majority of patients with above Milan HCC in our study population might be received LT from living donor. So, in our opinion, describing listing rules in Korea for HCC patients would be unnecessary in this study.

-sarcopenia is not used as a criteria used for matching, nevertheless it influences all the outcomes studied, especially renal function evolution

=> Unfortunately, data about sarcopenia have not been gathered in KOTRY registry. Instead, we matched patient's physical performance as KPS score.

-in the methods section there are two definitions lacking: biopsy proven rejection (any severity? T-cell and antibody-mediated?) and graft failure (needing retransplantation? bili above a certain threshold? recurrent ascites?)

=> Thanks for your opinion. We added those definitions as below.

<Page 4>

The primary endpoint of this study was BPR, which was defined any rejection reported from liver biopsy. The secondary endpoints were graft failure, which was defined as receiving retransplantation or patient death from graft dysfunction, and change in eGFR within 12 months after the matched time point.

-the fact that the reason for MMF non-utlilization or discontinuation is missing, as stated in the discussion. This is one of the most notable weaknesses of the study, since it can lead to confusion bias? For example, if diarrhea was a frequent reason, then some patients in the TAC alone group could have suffered of dehydration caused by the diarrhea, thus impacting the renal function.

=> Thanks for your good point of view. That is inevitable and important limitation of this registry study. However, we tried to minimize bias using novel matching procedure with many covariates including prior infectious complication before the matched time points and baseline renal function.

-there are typos in the text: page 2 line 80, all the citations in the Discussion section

=> We could not find any typo in page 2 line 80. Please specify which words are incorrect. Thanks.

-another limitation is the generalizability of the study: the cohort is pretty unique and difficult to compare to US or european LT cohorts since LDLT and HBV are overrepresented. This militation should be stated in the discussion section.

=> Thanks for your opinion. We added limitations in the discussion section as below.

<Page 10>

Also, our results could be limited for generalization because majority of study population of this study received LDLT and more than half were HBV infected patients unlike western population.

Round 2

Reviewer 1 Report

Thank you for your reply. 

Author Response

Thank you for your kind consideration.